# The Impact of Climate Variations on the Structure of Ground Beetle (Coleoptera: Carabidae) Assemblage in Forests and Wetlands

**Marina Kirichenko-Babko** [1,*], **Yaroslav Danko** [2], **Anna Musz-Pomorksa** [3],
**Marcin K. Widomski** [3] and **Roman Babko** [1]

1   Schmalhausen Institute of Zoology NAS of Ukraine, Department of Invertebrate Fauna and Systematics,
    B. Khmelnitsky 15, 01030 Kyiv, Ukraine; rbabko@ukr.net
2   Faculty of Natural Sciences and Geography, Sumy Makarenko State Pedagogical University, Romenskaja 87,
    40002 Sumy, Ukraine; yaroslavdanko@gmail.com
3   Environmental Engineering Faculty, Lublin University of Technology, Nadbystrzycka 40B, 20-618 Lublin,
    Poland, a.musz-pomorska@pollub.pl (A.M.-P.); m.widomski@pollub.pl (M.K.W.)
*   Correspondence: kirichenko@izan.kiev.ua; Tel.: +38-044-235-1070

**Abstract:** We studied the effect of climate variations on the structure of the assemblage of ground beetles (Coleoptera: Carabidae) in a wetland and surrounding watershed forest. We analyzed the changes in the structure of the assemblage of ground beetles provoked by a two-year dry period against the background of studies carried out during the two-year wet period. Aridization influenced the structure of the assemblage of ground beetles more in wetlands than in forests. It was shown that despite the stabilizing effect of the forest on the structure of assemblages of terrestrial arthropods, the two-year dry period had a negative impact on the assemblage of ground beetles in the studied area. The Simpson dominance index of 4.9 during the wet season increased to 7.2 during the drought period. Although the total number of species during the dry period did not significantly decrease in comparison with the wet period—from 30 to 27 species—changes occurred in the trophic structure: during the drought period, the number of predators decreased. It is concluded that the resistance of forest habitats to climate aridization is somewhat exaggerated and, very likely, the structure of the community of arthropods in forests will change significantly.

**Keywords:** humid forest; habitat quality; soil moisture; aridization; Carabidae; species distribution

## 1. Introduction

Actually observed climate changes, resulting from humans industrial, transport and agricultural activities triggering the green-house effect and changing precipitation volume and patterns, affect water availability [1,2] and occurrence of extreme weather-related events such as floods and heat waves [3–5]. Thus, the increased duration of dry periods between subsequent rainfall events causes threat of droughts in the areas affected by climate anomalies, usually expressed by decrease in precipitation during the warm period of the year [6]. Decreased precipitation and elevated temperatures are a serious threat to the water balance and biodiversity of natural ecosystems [7–9].

The most sensitive to increasing temperature and climate aridisation are those land areas where evolution took place under conditions of high water content, such as various types of wetlands, river valleys, streams and temporary streams. Wetlands cover approximately 6% of the of the Earth's land surface [10], and they are extremely vulnerable to the effects of climate change because they are very dependent on the water cycle. They are often found at the interface between terrestrial ecosystems, such as forests and grasslands, and water, such as rivers, lakes, estuaries, and oceans [11].

Wetlands and their biota are disappearing worldwide due to human activities, e.g., uncontrolled and unsustainable insufficient water resource management and increased water demand by growing urban populations [12–18]. In light of the above, global warming can be seen as a verdict against the conservation of biodiversity.

At the end of the 20th century and the beginning of the 21st century, the longest warming period in Eastern Europe took place over more than 120 years of systematic observations [19]. In Ukraine, from 1993 to 2010, the duration of the warm period increased by 4–10 d in Polesie and the forest-steppe and by 17–26 d in the steppe [20]. Under conditions of climate variations, with reduced rainfall at high temperatures, the distorted water balance of ecosystem will result in increased evapotranspiration quickly, leading to surface waters drying and a decrease in soil moisture in the range of plants root zones, subsequent reduction of water content in the unsaturated zone, and, finally, an increase in the water table level depth, thereby lowering the amount of retained water available for plants. The reduced water availability in the ecosystem in the form of surface and soil retention significantly endangers the environmental sustainability of the region by rearrangement of population distributions [21–23]. Special attention should be paid to support the natural forestation of ecosystems due to the significant ability of forests to intercept and retain precipitation water as well as limiting the ratio of soil, surface water and groundwater drying.

The increase in the duration of dry periods triggered by limited precipitation will obviously lead to a reduction in habitats for hygrophilous species, changing their populations' distribution and restructuring the ecosystem. It is not surprising that great attention is paid to studying the effect of temperature increases on individual biomes and their diversity on a global scale [24–28]. Thus, the influence of climate change on the reactions of animals from different taxonomic groups (birds, butterflies and amphibians, less often beetles) and the change in their geographic areal due to climatic changes are studied [29–33]. Terrestrial arthropods comprise most of the biodiversity in wetlands and include many rare and endangered wetland species [34–36]. It is quite possible that arthropods in the conditions of global warming will be practically deprived of refugia.

Among arthropods, ground beetles (Coleoptera: Carabidae) are considered to be useful environmental indicators that are important for understanding the patterns of changes in overall biodiversity [37]. Climatic variations have a significant impact on the level of soil moisture and, obviously, change the structure of their biological components. Ground beetles respond to changes in climatic conditions, but the speed and nature of the change in their assemblage are largely unknown.

The aim of our work is to establish the response of the assemblage of ground beetles to climate variations driven by aridization. We analyzed how climate variations affects the structure of the assemblage of ground beetles using the example of a local area.

## 2. Materials and Methods

### 2.1. Study Area

The study area is located in the temperate continental climate zone, and is a forested ravine–gully system surrounding the valley of a small river (Bytytsia river, right tributary of the Psel river, Dnieper basin) [38,39]. The studied valley of a branched ravine is situated in the woodland area—Vakalivschyna tract (wet oak forest, 150 m a.s.l., coordinates of the section of the ravine—51°02′353′′ N, 34°55′266′′ E and 51°02′249′′ N, 34°55′591′′ E), 22 km north of Sumy city (northeastern Ukraine).

In the early 90s of the 20th century, the valley of the ravine was swampy. During this period, the high humidity determined the microclimatic conditions in this ravine. The stream was maintained in spring during snowmelt and during rainy periods. The stream did not dry up during the year as it filled up due to infiltration of water from the forested watersheds along both slopes of the gully. Beginning in the 2000s, an increase in average temperatures in the region was accompanied by a decrease in the water content in the study area due to earlier melting of snow and a reduction of rainy

periods. As a result of the drought period, in summer, the stream dried up, resuming in spring during the snowmelt and in autumn during the rainy season.

This paper analyzed the data for two periods differing in climatic conditions: wet (1993, 1994) and drought (2009, 2010); the interval between which is 14 years. During this period in Ukraine, average temperatures steadily increased. The mean annual temperatures in the wet period were −4 °C in January and 25 °C in July; in the dry period, they were −3 °C in January and 31 °C in July [40]. The wet two-year period was characterized by a large amount of annual precipitation—from 1392 to 1452 mm, while the second period (also a two-year period) was dry, and compared to the first one, was characterized by half the amount of annual precipitation—from 610 to 584 mm [41] (Figure 1).

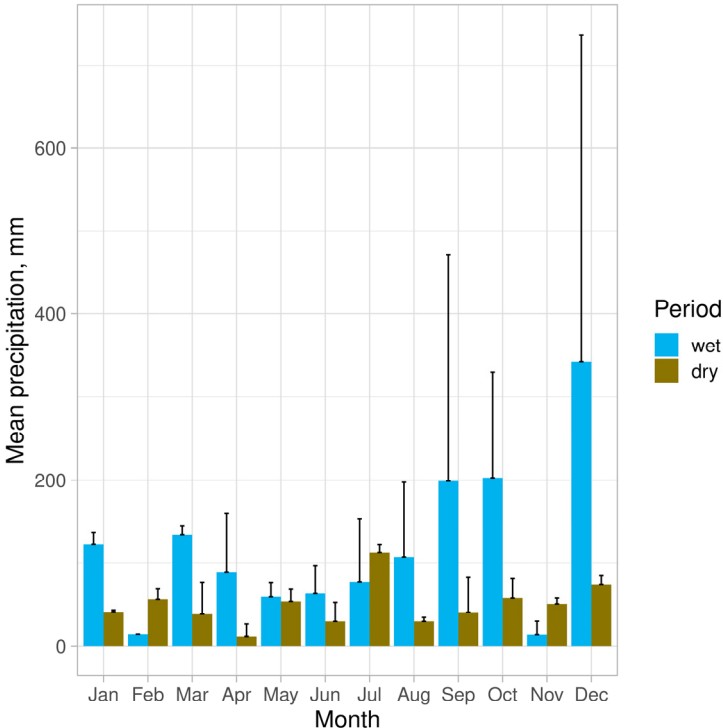

**Figure 1.** Average monthly precipitation during the two-year wet (1993–1994) and two-year drought (2009–2010) periods in the study area.

## 2.2. Sampling Design

The sampling was carried out in the same habitats (top, slopes and bottom of the ravine) during the vegetation period (from April to September) in two-year wet (1993, 1994) and in two-year dry (2009, 2010) climatic conditions. The traps were placed in three rows (along the transect) in the seven sections: at the top of both slopes, along the slopes, in the bottom of the ravine and the banks of the stream (Figure 2). Sampling was performed in 21 sites. At each sampling site, ten pitfall traps were placed at a distance of 10 m between traps. The trap is a polyethylene beaker 90 mm in diameter and 300 ml in volume with a solution of salt on the bottom. The traps were operated for two weeks every month, and samples were taken once a week.

In all sites of sampling, the soil moisture level was recorded. The values of humidity from 1 to 3 were considered as dry, from 4 to 7 as moist and from 8 to 10 as wet.

Carabid beetles were identified to the species level using the keys by Hůrka and Müller-Motzfeld [42,43]. Carabid species were classified according to their humidity preferences as hygrophilic, mesophilic and xerophilic according to Turin [44], and were divided to trophic groups following the literature [45–47].

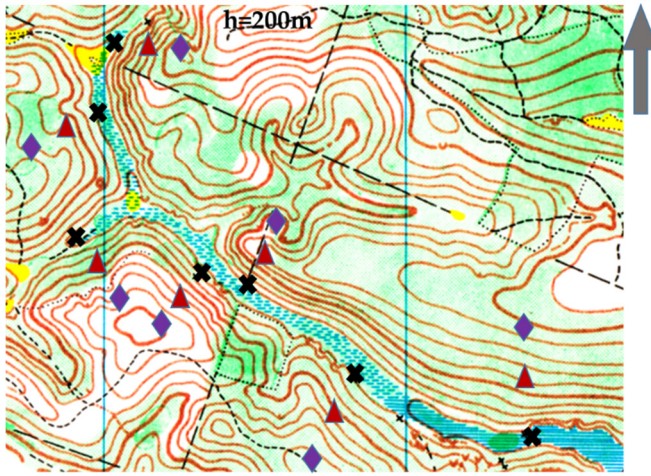

**Figure 2.** The studied branched ravine. The sampling sites and corresponding habitats are indicated as follows: the bottom of ravine or the banks of a temporary stream are marked with black crosses, the slope of the ravine is marked with red triangles, and the forest on the plakor is marked with purple diamonds. Scale 1:15,000. The arrow points to the north.

### 2.3. Statistical Analysis

Data were processed using R version 3.5.1 [48]. Principal component analysis (PCA) was performed with the PCA function in the FactoMineR package [49], and non-metric multidimensional scaling (nMDS) with metaMDS from vegan [50]. The quality of representation of the variables on the factor map was estimated by the $\cos^2$ index. A high cos2 indicates a good representation of the variable on the principal component and vice versa. Before analysis, the data were Hellinger transformed [51]. Tests with the rankindex function, which ranks correlations between dissimilarity indices and gradient separation from the vegan package, showed that the Kulczynski index is in the best accordance with the humidity gradient, and, thus, it was used. Figures were implemented using R packages ggplot2 [52], factoextra [53], ggrepel [54], directlabels [55].

Despite the small size of our data by computer standards, they can be classified as high dimensional, since the number of species exceeds the number of stations (for example, the matrix for the dry season has 21 columns and only 13 rows). Soil moisture is an important factor determining the quantitative development of ground beetle species. On the other hand, we can expect that a certain level of soil moisture corresponds to a certain level of quantitative development of a particular species. Therefore, we decided to build a model that would predict the expected soil moisture level based on the abundances of ground beetle species (see Table S1). For the construction of the models, we selected species (21 species) that were present in both wet and drought periods. As a training set, we used data on the abundance of ground beetles and soil moisture in the wet period, and as a test set, we used similar data for the drought period. Classical approaches such as least squares linear regression are not appropriate in these settings [56]. In the high-dimensional settings, more appropriate are dimension reduction methods, such as lasso and principal component regression (PCR). PCR belongs to unsupervised methods since response—humidity in our case—is not used to determine the principal component directions. On the other hand, lasso is the supervised dimension reduction method. We decided to use these two methods to see how well the results fit together. The source data and code are given in the Table S1: information computation S1. To build the lasso model, we used the glmnet R package. To build the PCR model, we used the pls R package.

The traditional community index, i.e., Simpson's dominance, is generally used to describe biological assemblages in order to infer ecological trends about the effects of disturbance [57,58]. The assemblage structure of ground beetles was estimated using Simpson's dominance index using the package Species Diversity & Richness [59]. The Sørensen index was used to compare the species composition of the ground beetle assemblage between the two periods [57]. Separation of

the two components of Bray–Curtis dissimilarity—balanced changes in abundance and abundance gradients—was performed according to Baselga [60].

## 3. Results

In total, 36 species of ground beetles were collected during the study periods (Table 1). In the wet period, 30 species of ground beetles were recorded in three habitats of the ravine (from 5 to 21 species by stations), and in the dry period, 27 species were recorded (from 4 to 13 species by stations). Of the 36 species registered in this area, 21 species were recorded in both periods. According to the Sørensen index, the species composition similarity under different climatic periods was 74%. The differences in the species composition in the studied periods consisted of the fact that nine species of ground beetles detected in the wet period were not recorded during the dry period. At the same time, during the dry period, six species were identified that are not recorded in the wet period (Table 1).

**Table 1.** Carabid beetle species caught in the study area, the percentage occurrence of each species in sites in the wet and drought weather conditions and information regarding their trophic requirement and humidity preference. Species grouped according to their hygro-preference.

| Species and Their Codes | | Occupancy of Sites (%) | | Trophic Requirement * |
|---|---|---|---|---|
| | | **Wet** | **Drought** | |
| Hygrophilous | | | | |
| *Abax parallelopipedus* Piller et Mitterpacher, 1783 | Ab.ater | 93 | 85 | o |
| *Abax parallelus* Duftschmid, 1812 | Ab.parallelus | 27 | 46 | o |
| *Agonum fuliginosum* Panzer, 1809 | Ag.fuliginos | 7 | 38 | c |
| *Agonum micans* Nicolai, 1822 | Ag.micans | 7 | - | c |
| *Agonum moestum* Duftschmid, 1812 | Ag.moestum | 27 | - | c |
| *Badister dorsiger* Duftschmid, 1812 | Ba.dorsiger | 7 | - | c |
| *Carabus granulatus* Linnaeus, 1758 | Ca.granulatus | 80 | 77 | c |
| *Carabus menetriesi* Faldermann, 1827 | Ca.menetriesi | - | 15 | c |
| *Cychrus caraboides* Linnaeus, 1758 | Cy.caraboides | - | 8 | c |
| *Elaphrus cupreus* Duftschmid, 1812 | El.cupreus | 40 | 23 | c |
| *Loricera pilicornis* Fabricius, 1775 | Lo.pilicornis | 13 | 31 | c |
| *Notiophilus palustris* Duftschmid, 1812 | No.palustris | 33 | 8 | c |
| *Oodes helopioides* Fabricius, 1792 | Oo.helopioides | 33 | 15 | c |
| *Oxypselaphus obscurum* Herbst, 1784 | Ox.obscurum | 7 | - | c |
| *Patrobus atrorufus* Stroem, 1768 | Pa.atrorufus | 7 | 15 | c |
| *Platynus assimile* Paykull, 1790 | Pl.assimile | 33 | 23 | c |
| *Pterostichus anthracinus* Illiger, 1798 | Pt.anthracinus | 7 | 15 | c |
| *Pterostichus diligens* Sturm, 1824 | Pt.diligens | 7 | 8 | c |
| *Pterostichus minor* Gyllenhal, 1827 | Pt.minor | 13 | 8 | c |
| *Pterostichus niger* Schaller, 1783 | Pt.niger | 7 | 38 | c |
| *Pterostichus nigrita* Paykull, 1790 | Pt.nigrita | 60 | 62 | c |
| *Pterostichus strenuus* Panzer, 1797 | Pt.strenuus | 7 | 15 | c |
| *Stomis pumicatus* Panzer, 1796 | St.pumicatus | 27 | 8 | c |
| Mesophilous | | | | |
| *Amara communis* Panzer, 1797 | Am.communis | 7 | - | g |
| *Anisodactylus signatus* Panzer, 1797 | An.signatus | 7 | - | g |
| *Asaphidion flavipes* Linnaeus, 1761 | As.flavipes | 7 | - | c |
| *Carabus cancellatus* Illiger, 1798 | Ca.cancellatus | 13 | - | c |
| *Carabus glabratus* Paykull, 1790 | Ca.glabratus | - | 8 | c |
| *Harpalus latus* Linnaeus, 1758 | Ha.latus | - | 15 | g |
| *Harpalus luteicornis* Duftschmid, 1812 | Ha.luteicornis | 13 | 8 | g |
| *Harpalus quadripunctatus* Dejean, 1829 | Ha.quadripun | 7 | 8 | g |
| *Pterostichus melanarius* Illiger, 1798 | Pt.melanarius | 27 | 54 | o |
| *Pterostichus oblongopunctatus* Fabricius, 1787 | Pt.oblongop | 40 | 54 | |
| Xerophilous | | | | |
| *Harpalus xanthopus winkleri* Schauberger, 1923 | Ha.winkleri | 13 | - | g |
| *Notiophilus germinyi* Fauvel in Grenier, 1863 | No.germinyi | - | 8 | c |
| *Pseudoophonus rufipes* De Geer, 1774 | Ps.rufipes | - | 23 | g |

Notes: Codes of the species are those used in Figures 5 and 6. * Abbreviations of trophic requirements of the carabid species: c—carnivorous, o—omnivorous, g—granivorous.

Information on the trophic structure of the assemblage of ground beetles in the studied area and their preferences to moisture is shown in Table 2. Table 3 shows the number of species of ground beetles recorded in the studied habitats of the ravine during periods with different climatic conditions.

**Table 2.** Characteristics of the species composition of the ground beetle assemblage in climatically different conditions by their trophic requirement and humidity preference.

| Species | Periods | | Shared Species | Total No. Species |
|---|---|---|---|---|
| | Wet | Dry | | |
| Hygro-preference/Feeding Group (in Adult): | | | | |
| **Hygrophilous:** | | **23** | | |
| Carnivorous | 19 | 16 | 14 | 21 |
| Omnivorous | 2 | 2 | 2 | 2 |
| Granivorous | 0 | 0 | 0 | 0 |
| **Mesophilous:** | | **10** | | |
| Carnivorous | 3 | 2 | 1 | 4 |
| Omnivorous | 1 | 1 | 1 | 1 |
| Granivorous | 4 | 3 | 2 | 5 |
| **Xerophilous:** | | **3** | | |
| Carnivorous | 0 | 1 | 0 | 1 |
| Omnivorous | 0 | 0 | 0 | 0 |
| Granivorous | 1 | 1 | 0 | 2 |
| Total no. species | 30 | 27 | 21 | 36 |

**Table 3.** Numbers of species in ravine habitats in climatically different conditions.

| Habitats | Periods | | Shared Species | Total No. Species |
|---|---|---|---|---|
| | Wet | Dry | | |
| Bottom of ravine | 21 | 22 | 17 | 27 |
| Slope of ravine | 12 | 11 | 6 | 17 |
| Forest on top slope | 15 | 9 | 8 | 16 |

The most abundant species in both periods were *Abax paralelepipedus* (37% of total catch), *Carabus granulatus* (9% of total catch), *Pterostichus oblongopunctatus* (8% of total catch), *Platynus assimile* (7.5% of total catch), *Pterostichus nigrita* (7% of total catch) and *Pterostichus melanarius* (5.4% of total catch).

In the humid period, at the bottom of the ravine, some species quantitatively prevailed, such as *Carabus granulatus*, *Pterostichus nigrita*, *Agonum moestum*, *Platynus assimile* and *Oodes helopioides*. In the forest in the watershed and on the slopes of the ravine, abundant species were *Abax parallelopipedus*, *Pterostichus oblongopunctatus*, *Pterostichus melanarius* and *Stomis pumicatus*. During the drought period, of the 27 species of ground beetles, the following species numerically prevailed at the bottom of the ravine: *C. granulatus*, *O. helopioides* (the same ones that prevailed in the wet period) and *Agonum fuliginosum*. In the forest in the watershed and on the slopes of the ravine, *A. parallelopipedus*, *P. oblongopunctatus*, and *P. melanarius* were abundant species during the humid period (Table 1).

During the humid period, four habitats were clearly distinguished within the studied territory: a forest on a plakor, slopes of a ravine covered with a forest, a swampy bottom of a ravine and a stream bank, clearly differ in the nMDS space (Figure 3a). The dry period significantly affected the quality of habitats in the studied area. In the dry period, due to the drying out of the stream at the bottom of the ravine, a decrease in humidity was observed. This decrease in humidity influenced both the species composition of ground beetle assemblage and their spatial distribution. In the dry period, the structure of the assemblage of ground beetles simplified, because the level of soil moisture in the studied area leveled. Additionally, two sites remained on the ordination plot: the bottom of the ravine and the forest in the watershed (Figure 3b).

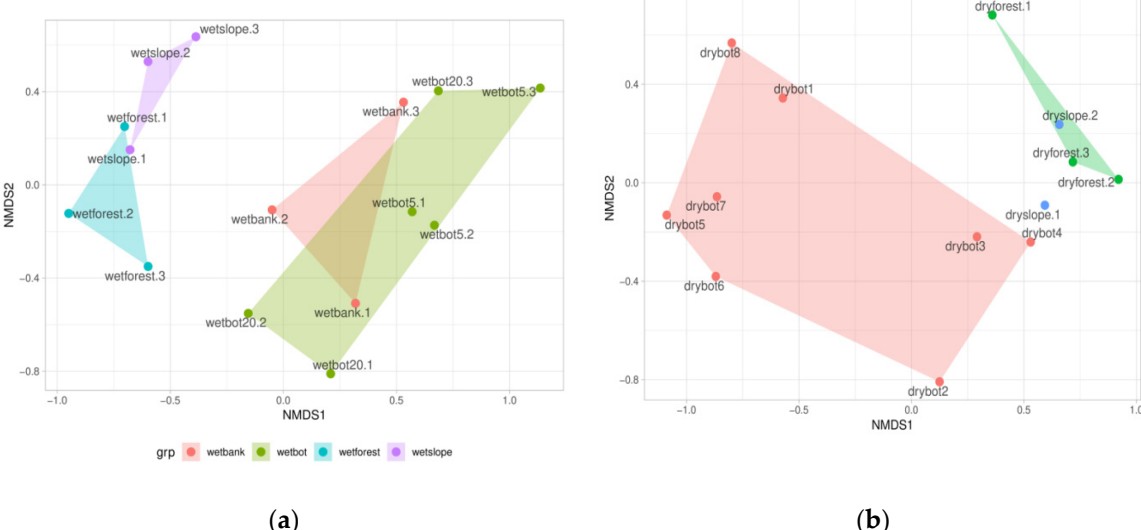

**Figure 3.** Non-metric multidimensional scaling plot of sampling sites of the forested ravine: (**a**) in the two-year humid period (stress $R^2$ = 0.119, goodness-of-fit: nonmetric 0.986, linear $R^2$ = 0.916); (**b**) in the two-year drought period (stress 0.096, goodness-of-fit: nonmetric $R^2$ = 0.991, linear $R^2$ = 0.953). Data were Hellinger transformed, and Kulczynski distances were used. Convex hulls show habitats. The names of the sites consist of three parts: first, "wet" or "dry" indicates the climatic conditions; the second part regards the habitat: "bot"—bottom of the ravine, "bank"—banks of the streams, "slope"—slope of the ravine, and "forest"—forest; and the third is a number indicating the sampling site.

The analysis for the entire studied period was performed including wet and dry years, examining how sites are distributed in the nMDS space on which the soil humidity gradient was superimposed (Figure 4). We analyzed the positioning of habitats (Figure 4a) and sites (Figure 4b) in the soil moisture gradient. Despite the dry period, the moisture level in the forest did not change significantly, while in the ravine, the stream dried up and the moisture content decreased from 6 to 3.5 (Figure 4a,b). During the wet period in the ravine, this indicator was kept in the range of from 6.5 to 8 (Figure 4a,b).

Climate variations have significantly less impact on the structure of the assemblage of ground beetles in the forest. Forest sites on the plakor and on the slopes of the ravine, both in wet and dry periods, were kept in a low moisture gradient of from 1.5 to 3.5. Since during the dry period the humidity under forest conditions did not change significantly, this ensured the stability of the structure of the assemblage of ground beetles in this habitat, which is confirmed by the localization of sites on the graph (Figure 4a,b).

Structural changes in assemblage under climate change conditions were assessed using Simpson's dominance index. At the bottom of the ravine, the value of the dominance index during the wet period was 7.9 and increased in the drought period to 11.5. In the forest, the value of the dominance index increased from 2.5 during the wet period to 3.2 during the drought period. In general, throughout the entire territory, the value of Simpson's dominance index in the wet period was 4.97, and it increased to 7.2 in the drought period.

Principal component analysis showed differences between the structure of the assemblage of ground beetles during wet and dry periods based on their soil moisture requirements. The results indicated that 22 carabid species in the humid period (Figure 5b) are represented by two groups depending on their relation to the level of moisture (Figure 5a). The cos2 plot (Figure 5b) demonstrates the quality of the variables.

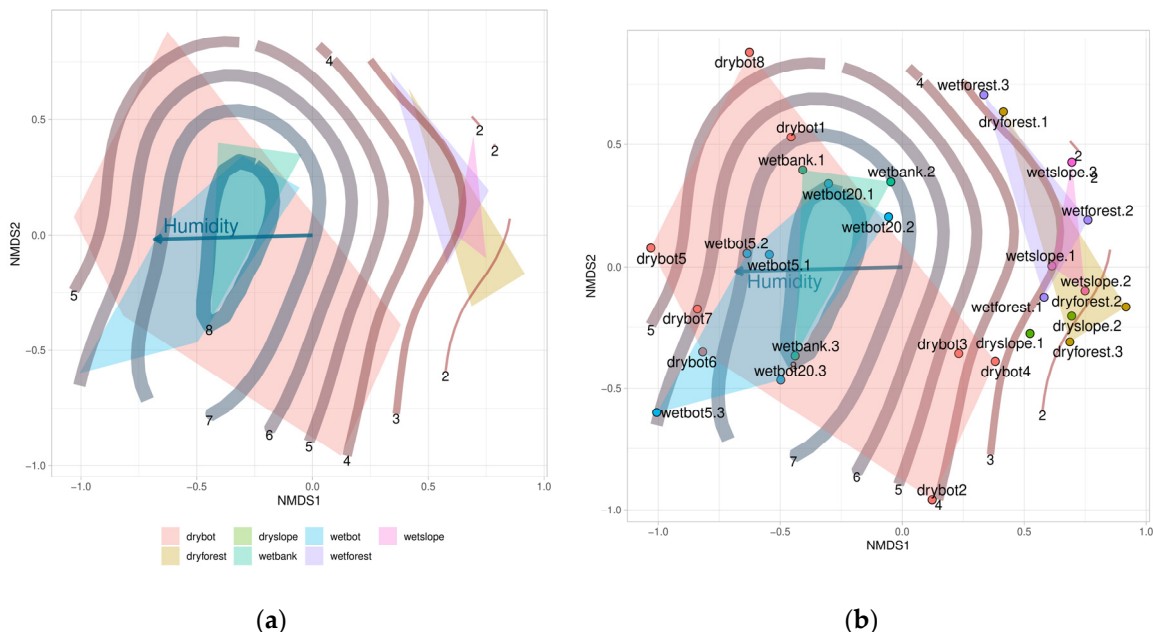

(**a**)          (**b**)

**Figure 4.** Non-metric multidimensional scaling plot of sampling sites in the forested ravine in humid and drought periods together with the gradient of humidity superimposed (stress: 0.154, goodness-of-fit: nonmetric $R^2 = 0.976$, linear $R^2 = 0.874$). (**a**) Only the habitat areas in the nMDS space; (**b**) the location of each site in the nMDS space. For details, see Figure 3.

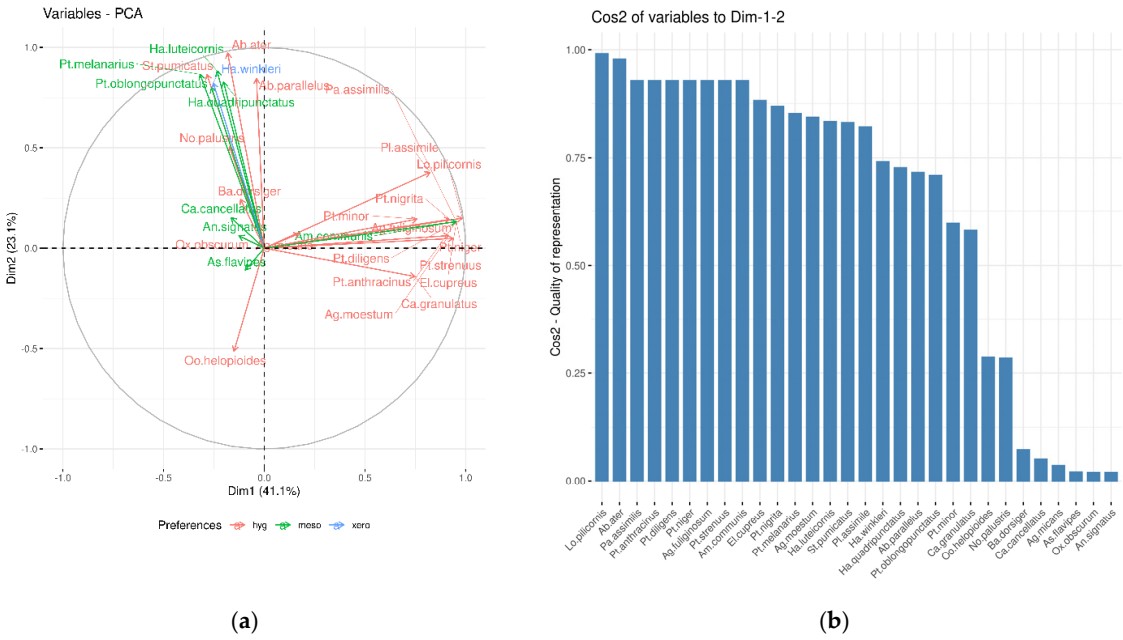

(**a**)          (**b**)

**Figure 5.** (**a**) PCA graph of standardized abundances of 30 ground beetles species in ravine habitats in the wet period. (**b**) Cos2 plot for ground beetles species in the wet period. Abbreviations: hyg—hygrophilous, meso—mesophilous and xero—xerophilous species. Codes of carabid species are given in Table 1.

Groups included the following significant species in the humid period (Figure 4a,b):

- Group 1: *Abax ater, Pterostichus melanarius, Harpalus luteicornis, Stomis pumicatus, Harpalus winkleri, Harpalus quadripunctatus, Abax parallelus,* and *Pterostichus oblongopunctatus*;

- Group 2: *Lorocera pilicornis, Patrobus assimilis, Pterostichus anthracinus, P. diligens, P. niger, Agonum fuliginosum, P. strenuus, Amara communis, Elaphrus cupreus, P. nigrita, A. moestum, Platynus assimile, P. minor*, and *Carabus granulatus*.

Eight species of the first group are practically limited to forest habitats: plakor and ravine slopes. Fourteen species of the second group are associated with the wet and shaded bottom of the ravine. Species of group 2 were associated with increased soil humidity in the bottom of ravines.

In the drought period, according to the results of PCA analysis, three groups of species were identified (Figure 6a); the plot cos2 indicated that only nine carabid species were significant (Figure 6b). Groups included the following significant species in this period:

- Group 1: *A. ater* and *P. oblongopunctatus*;
- Group 2: *P. anthracinus, P. assimile, N. germinyi, C. caraboides*, and *C. menetriesi*;
- Group 3: *P. nigrita* and *A. fuliginosum*.

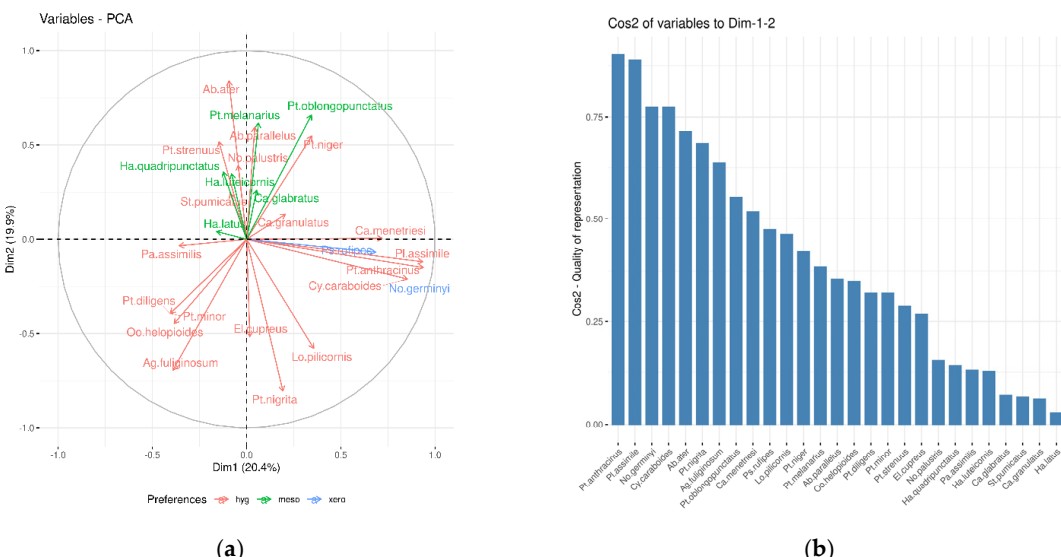

| (**a**) | (**b**) |

**Figure 6.** (**a**) PCA graph of standardized abundances of 27 ground beetles species in ravine habitats in the drought period. (**b**) Cos2 plot for ground beetles species in the drought period. Abbreviations as in Figure 5. Codes of carabid species are given in Table 1.

With the onset of the drought season, the number of significant species more than halved (Figure 6b). During this period, which began 14 years after the wet season, 6 of 22 significant species remained: *A. ater, P. assimile, P. oblongopunctatus, P. anthracinus, P. nigrita*, and *A. fuliginosum*. Three species—*A. communis, A. moestum,* and *H. winkleri*—out of 22 significant in the wet period were not recorded at all. At the same time, during the drought period, three species appear in the composition of a significant group that were absent in the humid period in this territory: *Carabus menetriesi, Cychrus caraboides*, and *Notiophilus germinyi*.

This is one of the reasons for the decrease in the number of significant species from 22 in the wet period to 9 in the drought period (Figure 5b and Figure 6b). Thus, significant group 1 species in the forest in the watershed and along the slopes decreased from eight species to two species during the drought period. At the bottom of the ravine, significant group 2 species decreased from 14 to 7 species in the drought years (groups 2 and 3). Changes in climatic conditions affected the number of hygrophilous species, which halved during the drought season (Table 4).

**Table 4.** Structure of groups of the significant species in wet and drought periods at the study territory. Groups 1–3 are indicated by PCA (Figures 5 and 6).

| Traits | Group 1 | Group 2 | Group 3 | Total Trait |
|---|---|---|---|---|
| Feeding group (in adult): | | | | |
| Carnivorous | 2/1 | 13/5 | 0/2 | 15/8 |
| Omnivorous | 3/1 | 0/0 | 0/0 | 3/1 |
| Granivorous | 3/0 | 1/0 | 0/0 | 4/0 |
| Hygro-preference: | | | | |
| Hygrophilous | 3/1 | 13/4 | 0/2 | 16/7 |
| Mesophilous | 4/1 | 1/1 | 0/0 | 5/1 |
| Xerophilous | 1/0 | 0/1 | 0/0 | 1/1 |
| Total in each group | 8/2 | 14/5 | 0/2 | 22/9 |

As a result of using the lasso model, four species with non-zero coefficients remained: *N. palustris*, *C. granulatus*, *P. melanarius*, and *P. oblongopunctatus*. Applying this model, in which these four species were used as predictors, we obtained the expected moisture values in the drought period to the test data (Table 5, Figure S1). The first two principal components explained 97% of the variance. Using the resulting model with two principal components to predict soil moisture in the drought period, we obtained the results shown in the Table 5 (Figure S1).

**Table 5.** Soil moisture in the drought period based on measurements and modeling results (lasso and PCR).

| Site | Humidity in Dry Period | Predicted by Lasso | Predicted by PCR |
|---|---|---|---|
| drybot1 | 4 | 6 | 7 |
| drybot2 | 4 | 6 | 7 |
| drybot3 | 4 | 6 | 7 |
| drybot4 | 4 | 6 | 6 |
| drybot5 | 4 | 6 | 7 |
| drybot6 | 4 | 6 | 7 |
| drybot7 | 5 | 7 | 7 |
| drybot8 | 5 | 6 | 7 |
| dryforest1 | 2 | 6 | 6 |
| dryforest2 | 2 | 5 | 5 |
| dryforest3 | 2 | 4 | 5 |
| dryslope1 | 2 | 6 | 5 |
| dryslope2 | 2 | 6 | 5 |

Notes: Abbreviations of sites as in Figure 3.

## 4. Discussion

It is known that forests play a stabilizing role, since soil moisture is more stable under the forest canopy, which determines the high species richness of ground beetles and their spatial distribution [61,62]. Climate variations and an increase in temperature lead to a decrease in the water content in the upper soil layers, which, combined with a decrease in precipitation, affects the quality of habitats [63]. It was also shown that temperature and moisture of soil plays an important role in the successful development of eggs and soil-dwelling larvae and, therefore, in the dynamics of carabid populations [64]. The low water content in the soil leads to its compaction, which makes it difficult for many soil animals to move. Ultimately, changes in humidity have a significant impact on the distribution of ground beetles and other epigeic arthropods and the structure of their assemblages [65]. A decrease in humidity often leads to a decrease in the abundance and diversity of arthropods [66,67]. Stenotopic and hygrophilous species are usually the first to respond to a decrease in soil moisture, and species with wide ecological plasticity and xerophilous gain a certain advantage.

According to the results of our research, even a relatively short dry period led to visible changes in the structure of ground beetle assemblages. In drought years, the number of microhabitats

decreased (Figure 3b). Considering that climate variation affects ground beetles primarily through a decrease in soil moisture, an important aspect is the ratio of groups of species in terms of their hygro-preferences, as well as the characteristics of the trophic structure of the significant groups of species (Table 4). The drought period also affected the trophic structure of significant groups of ground beetles. The number of predators during the drought period decreased to 5 from 15 species present during the wet period. During the drought period, populations of two predators (*A. moestum* and *S. pumicatus*) and two granivorous species (*A. communis* and *H. winkleri*) were not recorded. During the drought period, in the composition of assemblage, three species of predators—*C. menetriesi*, *C. caraboides*, and *N. germinyi*—appeared in the structure of a significant group, which had not been observed before.

According to the results of our study, carried out 14 years after the wet period, the total number of recorded species in the study area changed slightly: from 30 to 27 species. It is generally accepted that under stress conditions, under the influence of negative factors, dominance increases [68]. Increased dominance indicates that the ecosystem is under stress. Our studies have shown that even in relatively stable forest ecosystems, changes in the structure of ground beetle assemblage are quite noticeable under the influence of warming.

Ultimately, our results confirm that the complexity of the structure of ground beetle assemblages correlates with the number of microhabitats available, and the simplification of conditions at the landscape leads to a decrease in the amount of available food resources for both predators and granivorous species [69]. Omniphagous species are less sensitive to such changes. It is known that representatives of higher trophic levels (carnivorous) react to the amount of precipitation [70]. The low number of granivorous taxa in our studies is explained by the fact that they predominate in open habitats [71], and they are known to also be sensitive to moisture reduction [72]. At the same time, seed consumption increases with increasing temperature among granivorous taxa [73].

It can be seen that the soil moisture values expected according to both models significantly exceed the observed ones. This result can be understood in two ways. The first interpretation is that soil moisture is not as important for ground beetles as is commonly believed. The second interpretation is that soil moisture, as it is generally accepted, is an important component of the ground beetle niche. However, the reaction of the assemblage of ground beetles to changes in soil moisture is not instantaneous; it is slowed down by evolutionary acquired mechanisms that make it possible to tolerate certain fluctuations in moisture levels. Therefore, judging by the current state of development of populations of ground beetles, we can conclude that the level of soil moisture is higher than it actually is, and this is what both our models show. In our opinion, the second interpretation is more likely.

As pointed out by Baselga [60], the Bray–Curtis index of dissimilarity is insensitive to some important differences in species abundance patterns. The first such situation is that the abundance of some species declines from site 1 to site 2 in the same magnitude that the abundance of other species increases from site 1 to site 2. This pattern is called balanced variation in species abundances. The second situation, the abundance gradient, is the observation that the abundance of all species equally declines (or increases) from site 1 to site 2. The Bray–Curtis index can take on the same value in these different situations. Obviously, there are intermediate states between these extreme situations. One possible solution is to subdivide the Bray–Curtis index into two components. In this case, the ratio of these components allows us to conclude which of the mentioned patterns prevails. The values of the Bray–Curtis index, subdivided into these components, for sites in the wet and dry periods are shown in Table 6. As can be seen, in more than half of the cases, the main contribution to the Bray–Curtis index is made by balance variation. This fact confirms our conclusion that due to aridization of conditions, the abundances of some species (stenotopic) decreases while that of others (eurytopic) increases. In other cases, the contribution of both components is equal, or gradient differences between sites prevail, which can be considered as evidence of a general deterioration of conditions for ground beetles in these habitats.

**Table 6.** Bray–Curtis dissimilarity between sites in drought and wet periods divided into components: balanced changes in abundance and abundance gradients.

| Wet Period | Drought Period | Bray | Balanced (B) | Gradient (G) | B to G Ratio |
|---|---|---|---|---|---|
| bank-3 | bot8 | 0.9395 | 0.8890 | 0.0505 | B |
| bank-2 | bot7 | 0.7910 | 0.7409 | 0.0501 | B |
| forest-2 | forest-2 | 0.4577 | 0.3750 | 0.0827 | B |
| bot5-1 | bot4 | 0.8188 | 0.5455 | 0.2732 | B > G |
| bot20-3 | bot3 | 0.7362 | 0.4545 | 0.2817 | B > G |
| bot20-2 | bot2 | 0.6469 | 0.4791 | 0.1677 | B > G |
| bot5-2 | bot5 | 0.5228 | 0.3823 | 0.1404 | B > G |
| forest-1 | forest-1 | 0.8846 | 0.4286 | 0.4559 | B ≈ G |
| forest-3 | forest-3 | 0.5969 | 0.2964 | 0.3006 | B ≈ G |
| slope-2 | slope-2 | 0.3476 | 0.1667 | 0.1809 | B ≈ G |
| slope-1 | slope-1 | 0.6403 | 0.1249 | 0.5154 | B < G |
| bank-1 | bot6 | 0.8991 | 0.0688 | 0.8304 | G |
| bot20-1 | bot1 | 0.7373 | 0 | 0.7373 | G |

Notes: bray—Bray–Curtis dissimilarity, balanced—balanced variation in abundances, gradient—abundance gradients.

It should be emphasized that our studies demonstrate the difficulty in interpreting the obtained results associated with the initial stage of restructuring of the ground beetle assemblage. We state that climate variations towards warming and aridity in the studied local area have led to a decrease in the number of trophic groups and the number of species included in them. Thus, in drought years, in comparison with the humid period, the group of significant species decreased by almost 2.5 times.

## 5. Conclusions

The results obtained allow us to assert that even moderate aridization causes noticeable changes in the structure of communities of terrestrial arthropods. Changes in the structure of the community of ground beetles were manifested in an increase in the level of dominance, changes in the composition of trophic groups and the number of hygrophilous species. Aridization mostly affected the structure of assemblages of ground beetles in humid habitats at the bottom of the ravine; however, under forest conditions, disturbances in the structure of the ground beetle community turned out to be more significant than expected.

Of course, if the trend towards higher temperatures and lengthening warm periods of the year continues, then the resistance of ecosystems to stress will decrease. The structure of ground beetle communities can be a convenient indicator to predict the degree and rate of decline in the stability of forest ecosystems.

**Supplementary Materials:** The following are available online at http://www.mdpi.com/1999-4907/11/10/1074/s1, Figure S1: Humidity models; Table S1: Data of species, information computation S1: code for building models.

**Author Contributions:** Conceptualization, M.K.-B., R.B.; methodology, M.K.-B., R.B.; software and validation Y.D., R.B.; model building, R-coding, Y.D.; formal analysis, M.K.-B., R.B.; investigation, M.K.-B., R.B.; resources, M.K.-B., R.B.; data curation, M.K.-B., R.B.; writing—original draft preparation, M.K.-B., R.B., and Y.D.; writing—review and editing, M.K.-B., R.B., Y.D., A.M.-P. and M.K.W.; visualization, M.K.-B., R.B., and Y.D.; supervision, M.K.-B., R.B. All authors have read and agreed to the published version of the manuscript.

**Funding:** This research received no external funding.

**Acknowledgments:** We thank the three reviewers for useful suggestions that helped to improve this paper.

**Conflicts of Interest:** The authors declare no conflict of interest.

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
