# Peer review of "The Impact of Climate Variations on the Structure of Ground Beetle (Coleoptera: Carabidae) Assemblage in Forests and Wetlands"

_forests, doi:10.3390/f11101074_

Round 1

Reviewer 1 Report

Dear authors,

I think your study is interesting, and worthwhile to publish in Forest. The study is well-presented in the introduction and methods could be easily replicated as many details are given on location, sampling and classification of beetles (trophy, ecological preferences). Nevertheless, I have several concerns related to the conceptualization of the paper, concluding remarks and data treatment, which I point below:

1) doubts on the use of 'climate change' in this particular situation

I am nor certain that the temporal window of 15 years between sampling dates can express a truly climate change or a variation in sampling periods. This issue has to be better addressed. In Figure 1, you present a variation on monthly rainfall patterns and a regression line, however, there are no analyses on the significance of the difference between the two periods. That is, no analyses are showing the significance of temperature and rainfall variation on the specific sampling season or between the two periods (only R2 of the monthly variation and not between periods). This should be addressed and explained. 

2) partioning of variation

Also, as other predictor variables (habitats, soil) were not assessed, I have doubts if the beetles are responding solely to temperature and rainfall variations. 

While adding new data around this problem (other variables) will certainly not possible, I think the paper must acknowledge all limitations clearly in the discussion. The limitations refer to: the temporal window (is 15 years of interval enough in Ukraine to assess climate change impacts?), the influence of other environmental variables, the influence of different sampling teams and expertise (this is not clear in the paper). This should be properly addressed and solutions for future research should be given. 

Given this two issues, if you cannot validate properly that we are facing the effects of climate change, I would propose a more careful title, maybe removing climate change and replace by warming or climate variation. This should also be revised across the text.

3) Sampling and data treatment

Sampling: was the sampling and identification made by the same team in the different periods?

nMDS: I would like to have information on the stress of the nMDS (2 dimensions)

4) Results and discussion

Table 1: there several hygrophyllous species that present more occupancy in the dry period than in the wet period (e.g. Ab. parallelus, Ag. fuliginosus) and one xerophyllous species that prefer the wet period. I found this as a contradictory result, and maybe I miss the discussion, but this was not fully addressed.

lines 289-29: the sentence "If the warming trend continues, these changes will be even more pronounced" is speculative, so remove as no predictive analyses were done, as far as I understood.

Minor comments:

Figure 1 - x-axis title month, not mounth

Figure 1: write extensively the word "Poly."

Figure 1: include the significance of R2

Table 1 correct comma after classifiers in some species

Reviewer 2 Report

General comments

The article addresses an interesting and little-documented topic, the effect of climate change on the ground beetle community. The topic is very interesting and, although the experimental design is not entirely clear, the data collected has great potential. However, the authors need to make an effort to better organize the document. The introduction is well written but remains general, it is necessary a better explanation of the hypotheses and the expected responses from this important group of insects.

Regarding the methods, the data analysis should be rewritten. The authors dedicate themselves to listing the used methods, but they do not explain how they used their variables. Some of the analyses have not been included in this section. Furthermore, the relationship between the analysis and the objectives is not clear. It is necessary to complement analyzes to better support the discussion and conclusions. Unfortunately, many points in the discussion are based on the authors' speculation but are not supported by the analysis. The results are confusing, many times the authors included information that should be included in methods, and many times the results are discussed. It is necessary to make an effort to generate a clearer structure in this section.

The authors propose an interesting speculation but there are not based on the analysis, many of these speculations are based on descriptive data. The conclusion is not sufficiently supported by the results and discussion.

In general, the document should be better structured, there should be a better connection among the different sections (introduction, methods, results, and discussion), in a way that the guiding thread of the document is appreciated.

Specific comments

Line 50-54: this paragraph is not related to the topic addressed in the paper, what is the interest of knowing the effect of tree cover on urban areas, your study does not address this issue.

Line 55-64: please integrate this paragraph to the first part, next to line 37.

Line 65: limit the use of expressions like "So"

Line 101-103: Delete this expression with reference to Figures. Just cite the figure (Figure 1)

Line 105-106: Better explain what you mean by “Poly”. Eliminate the legends from the figure and explain in the figure caption.

Line 109: It is necessary to explain how the authors selected the 21 sampling sites. How the authors ensured to have equal representation in each of the habitats?

Line 110: It is necessary to explain how the pitfall traps were located; in a transect, in a grid?. How that location was defined in relation to the slope and other variables that could affect the data?.

Line 112: Did the traps stay open for 24 months? It needs to be clarified.

Line 112-114: It is better to avoid expressions such as: “the soil moisture level was recorded”, it is better to put directly what and how it was measured. It is not clear what they mean by humidity. units. The authors propose a numerical scale from 1 to 10, but the unit is not known, then they transform these units into categories.

Line 115: Delete expression "The obtained material was sorted in the laboratory"

Line 115-118: Although the citation format is number, in some cases such as those cited in this paragraph, it is necessary to cite the name of the authors.

Line 121-122: It is not necessary to justify the use of the R programming environment. Delete this sentence.

Line 122-124: The authors presented the used tests but they did not explain how they used them and what they are using for.

Line 125: Was the data transformed? It does not make sense, in line 108 the authors proposes that the data were standardized, why the data were standardized and transformed? It is important that the authors taken into account the impact of standardization on the results. Standardization eliminates the effect of dominant species, giving equal importance to rare species. According to what he raises in the study, it is important to see these differences.

Line 125-126: again, the authors mention a “Rank test” analysis but they do not explain why they implement it or how it is implemented.

Line 126-127: It is better figures instead of Plots, and implemented instead of produced

Line 128-130: The explanation is not clear. The authors suggest that the matrix has 21 columns by 13 rows, however, according to the methodology it should have 21 locations, therefore 21 rows. I understand that the matrix should be 36 columns by 21 rows. In any case, I do not consider that this difference is enough to say that there is a high dimensionality.

Line 138-141: Again the authors describe the method used but not how they implement it. At the site level? At the habitat level?

Line 146: The Sorensen calculation does not appear in methods, it is necessary to clarify how it was implemented and at what level. It would be interesting to use the partition of the betadiversity (see Baselga et al. 2013 - Methods in Ecology and Evolution) to assess the process behind the loss. It seems to me that the information on beta diversity should be better developed, for example, using abundance data and evaluating the beta diversity of the same habitats between dry and wet years, this information would help to understand which habitats have lost more species.

Line 161: Cite table 1

Line 162-165: It is not necessary to quote the tables in this way. The authors are required to describe trends. It would be interesting to test whether the structure, due to its affinity to humidity, changes between periods and if this change is different between habitats. Table 2 mixes two categories that are different, it is necessary to separate or make a table that shows the interaction of the two variables (trophic level and humidity affinity). I think this information is very important and much of the discussion is done on this, unfortunately this has not been tested and much is only assumptions.

Line 179-181: In the methods, it is necessary to clarify about these habitats and how they were established.

Line 181-182: This sentence; “The dry period significantly affected the quality of habitats in the studied area” is not supported by any information presented in the document, it is an appreciation of the authors.

Line 182-183: It is not clear what the authors are based on to make this statement, it is necessary to explain the results.

Line 183-184: again because the authors state that the structure was simplified, this can have many interpretations. Do you mean that sites in the same habitat became more similar?

Line 199: keep the two-year expression used throughout the manuscript and not biennial

Line 209-246: This entire section is difficult to understand because it has not been explained in methods.

Line 244-246: This sentence corresponds to the discussion

Line 249-262: this part corresponds to the methods

Line 266-269: This line does not correspond to the discussion

Line 283-284: The authors propose that micro-habitats disappear, and base this statement on the fact that there is not a polygon in the biplot that represents certain micro-habitats. However, that does not necessarily mean that micro-habitats disappear, it may be that the community is more similar so there are no differences between sites, and therefore there is no polygon.

Line 284-289: Results

Line 300-304: This paragraph corresponds to the results

Line 290-292: Results, how do you know if this difference is significant and it is not a change due to the reduction of the sample size (number of individuals)?

Reviewer 3 Report

Review’s comments

Manuscript Number: forests-922722

General comments

The manuscript entitled “The Impact of Climate Change on the Structure of the Ground Beetles Assemblage in Forest and Wetland” dealt with interesting topic in aspects of climate change. Probably, aridization greatly affects insects in the level of population or community. In spite of this virtue, the manuscript improved in aspects of clarity. The sources of data, collection date and experimental design are not clear. 

1) Basic information for sampling periods, locations and sampling design is missing in your manuscript. When and where did you collect ground beetles? Sampling year and exact location (e.g. GPS coordinate) are missing in Materials and Methods. Because sampling methods is missing in your manuscript, sampling design of your experiments is not clear. Probably map of sampling locations and diagram for pitfall arrangement is better to be included. 
2) All of beetle data were presented in the species number or proportion among total catch. The number of individuals collected is essential information for readers to understand your works.
3) Statistic for ordination studies were missing. For example, NMDS requires stress, instability and Monte Carlo permutation tests. Statistical tests for your ordinations should be included in the manuscript.
4) Title of the manuscript is needed to be corrected for your focus. This study mainly dealt with aridization rather than climate change.

Specific comments

line 175, Table 2. Different criteria for beetles was mixing in the table. Correction like table 4 is necessary. 

Round 2

Reviewer 1 Report

Dear authors,

Thanks for your careful edition of the manuscript, and efforts on improving it according to suggestions. I am glad to inform you that the manuscript is suitable for publication in the current form.

Author Response

Dear Reviewer(1), we are grateful for the comments and recommendations that allowed us to improve this manuscript.

Reviewer 3 Report

In Fig. 2., scale bar for distance is missing. Please add it to map. I thought that colors and lines in the map is not necessary.

Author Response

Dear Reviewer (3),

We are grateful for the comments and recommendations that allowed us to improve this manuscript.